# Air pollution-induced proteomic alterations increase the risk of child respiratory infections

Early life air pollution exposure may play a role in development of respiratory infections, but underlying mechanisms are still not understood. We utilized data from two independent prospective birth cohorts to investigate the influence of prenatal and postnatal ambient air pollution exposure of $PM_{2.5}$, $PM_{10}$ and $NO_2$ on maternal and child proteomic profiles and the risk of daily diary-registered common infections age 0-3 years in the Danish $COPSAC_{2010}$ (n = 613) and pneumonia, croup and bronchitis age 1-2 years in the Swedish EMIL (n = 101). A supervised sparse partial least square model generated proteomic fingerprints of air pollution analyzed against infection outcomes using Quasi-Poisson and logistic regression models, respectively. Here we demonstrated that prenatal ambient air pollution exposure was associated with altered maternal proteomic profile with significant downregulation of the AXIN1 protein. The prenatal air pollution proteomic fingerprints related to a significantly higher risk of total number of infections, cold, pneumonia and fever episodes in $COPSAC_{2010}$ and similar postnatal air pollution proteomic fingerprints related to a significantly higher risk of respiratory infections in EMIL. Higher AXIN1 protein levels associated with significantly decreased risks of total number of infections, cold, pneumonia, tonsillitis and fever episodes, and asthma risk in $COPSAC_{2010}$ and a significantly decreased risk of respiratory infections in EMIL suggesting a protective effect of this specific protein in both cohorts. This study of two prospective birth cohorts demonstrates ambient air pollution alterations in the maternal and child's proteomic profiles that associates with respiratory infection risk suggesting the AXIN1 protein as a potential target for respiratory infection and asthma prevention in childhood.

Ambient air pollution exposure in early life is a well-known risk factor for development of asthma, allergy, and lung function deficits throughout childhood[1–5]. More specifically, studies have identified an increased exposure to fine particulate matter <2.5 μm ($PM_{2.5}$) during pregnancy to possess a risk for disturbances in alveolarization, pulmonary immune differentiation, and neonatal immunity[6,7], suggesting air pollution increases the risk of infant mortality through an early life respiratory infection load[8]. In a WHO Global Health Observatory study

of preschool-aged children, ambient air pollution-induced lower respiratory tract infections (LRTIs) accounted for around 20% of losses in life expectancy, equivalent to 21.5 million years of life lost due to air pollution exposure in 2015, emphasizing air pollution as an important factor for childhood health[9].

The link between prenatal ambient air pollution exposure and risk of respiratory tract infections in childhood is conflicting[10,11], where a large pregnancy study of 17,533 children in the Norwegian Mother and

✉e-mail: nicklas.brustad@dbac.dk

Child Cohort Study (MoBa) showed no association until age 18 months[12]. However, a previous study across 10 European birth cohorts ($n = 16,059$) linked air pollution exposure at birth to risk of pneumonia in early childhood[13] and recently (2024), a large study ($n = 224,214$) found ambient $PM_{2.5}$ exposure in early childhood to be associated with increased risk of respiratory infections in pre-school aged children as well[14]. Finally, air pollution exposure in early life has been associated with alterations in the inflammatory proteomic profile, showing, e.g., downregulation of IL8[15]. In another study, overexpression of IL8 in lung epithelium has been shown to benefit lung immunity to bacterial infection[16]. However, these mechanistic pathways are yet to be investigated in relation to childhood respiratory infection risk.

To date, no prospective mother-child cohort study utilizing unique information on daily infection symptoms and diagnosis load during the first 3 years of life has investigated the relationship between pre- and postnatal ambient air pollution exposure, maternal and child inflammatory proteomic profiling, and risk of common infections in childhood. We hypothesize that air pollution exposure in the pre- and postnatal period changes the inflammatory proteomic profile of the pregnant mother and newborn child, with associations to later respiratory infection risk.

## Results

In the COPSAC$_{2010}$ cohort, 613 (88%) children had available diary data on infection status from birth until age 3 years, information on prenatal air pollution exposure, and also had mothers having an inflammatory proteomic profiling done in pregnancy week 24 (Supplementary Fig 2). We observed a mean (SD) of 16.3 (8.3) of total number of infection episodes (Supplementary Table 2). The most frequent infection type was cold with a mean (SD) of 12.2 (8.0) episodes per child (Fig. 1). There were no differences in baseline characteristics including parental asthma status and pre-, peri-, and postnatal exposures between children having versus not having available infection diary data as previously reported[17]. All three air pollution measurements of $PM_{2.5}$, $PM_{10}$, and $NO_2$ were highly correlated.

### Prenatal air pollution exposure vs the risk of early childhood infections

We found some evidence that the children whose mothers were most exposed to air pollution measured as concentrations of $PM_{10}$ throughout pregnancy had an increased risk of gastrointestinal infections age 0–3 years, adjusted IRR (95% CI): 1.13 (1.00–1.26), $p = 0.046$. There was no association with other infection types or the overall infection burden for $PM_{2.5}$, $PM_{10}$, and $NO_2$ (Supplementary Fig 3). All analyses were adjusted for gestational age, furred pets during the first year, maternal education and income, time to daycare start, number of older siblings, alcohol use, antibiotic use, and smoking during pregnancy, delivery mode, child hospitalization at birth, and birth season.

### Prenatal air pollution exposure and the maternal proteomic profile

Air pollution exposure during pregnancy associated with levels of several maternal inflammatory proteins for both $PM_{2.5}$, $PM_{10}$ and $NO_2$ as illustrated by the Volcano plots in Fig. 2. For $PM_{2.5}$ exposure AXIN1, CXCL6 and IL8 were significantly downregulated, for $PM_{10}$ exposure AXIN1, IL8, SIRT2, ST1A1 and STAMBP were significantly downregulated whereas for $NO_2$ exposure IL33 was significantly upregulated (Fig. 2). The false discovery rate (FDR) adjusted p-values are outlined in the Volcano plots illustrated in Supplementary Fig 4 showing similar significant associations with downregulation of AXIN1 and IL8 from PM2.5, AXIN1 from PM10 and upregulation of IL33 from NO2 exposure.

We performed a PCA of the 92 maternal inflammatory proteins where principal component 1 (PC1) explained 30.9% of the variation and PC2 6.5% as illustrated by the PCA plot in Fig. 3A. Dividing mothers into groups of high vs low ambient air pollution exposure divided by the medians showed that high $PM_{2.5}$, $PM_{10}$ and $NO_2$ associated with PC2 but not PC1 (Fig. 3B). A high PC2 score was characterized by elevated levels of AXIN1, STAMBP, SIRT2, ST1A1 and IL8 among others, which were also the proteins downregulated by exposure to $PM_{2.5}$ and

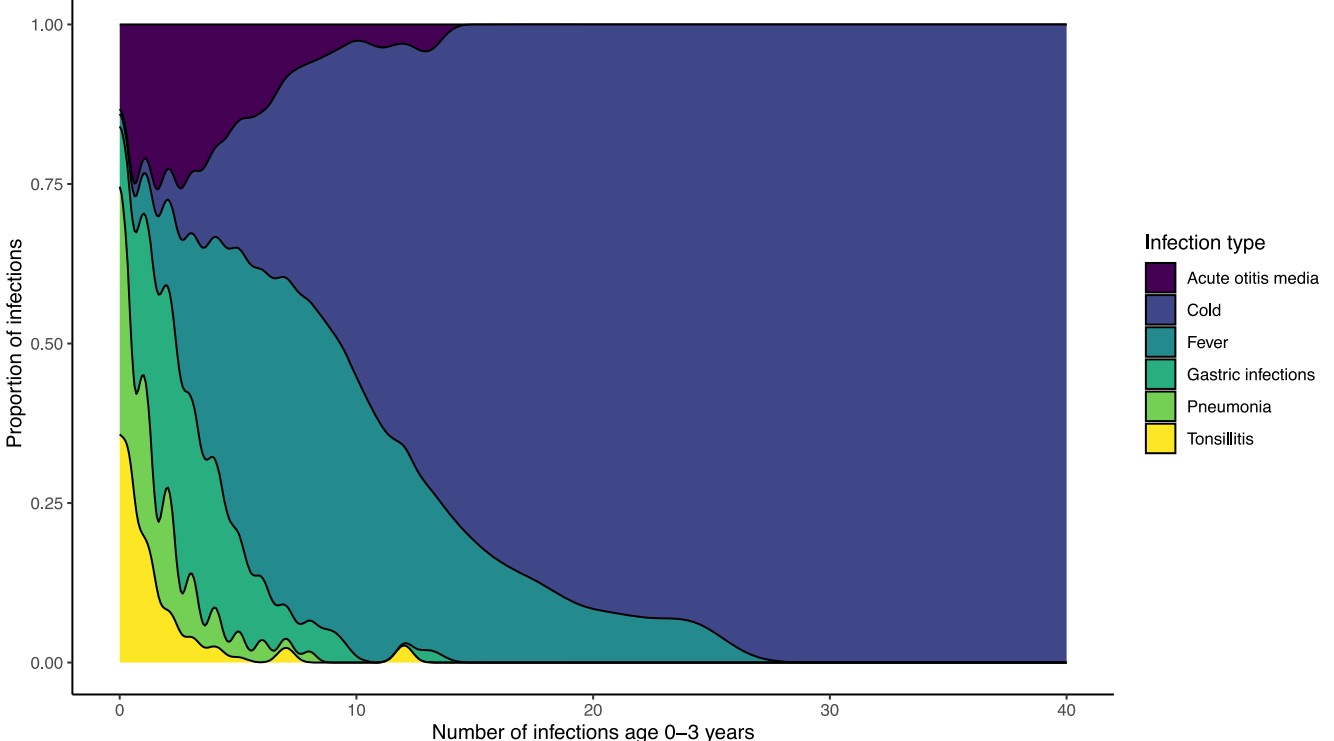

**Fig. 1 | Density plot.** Density plot showing the proportion of infection types age 0–3 years in the COPSAC$_{2010}$ cohort.

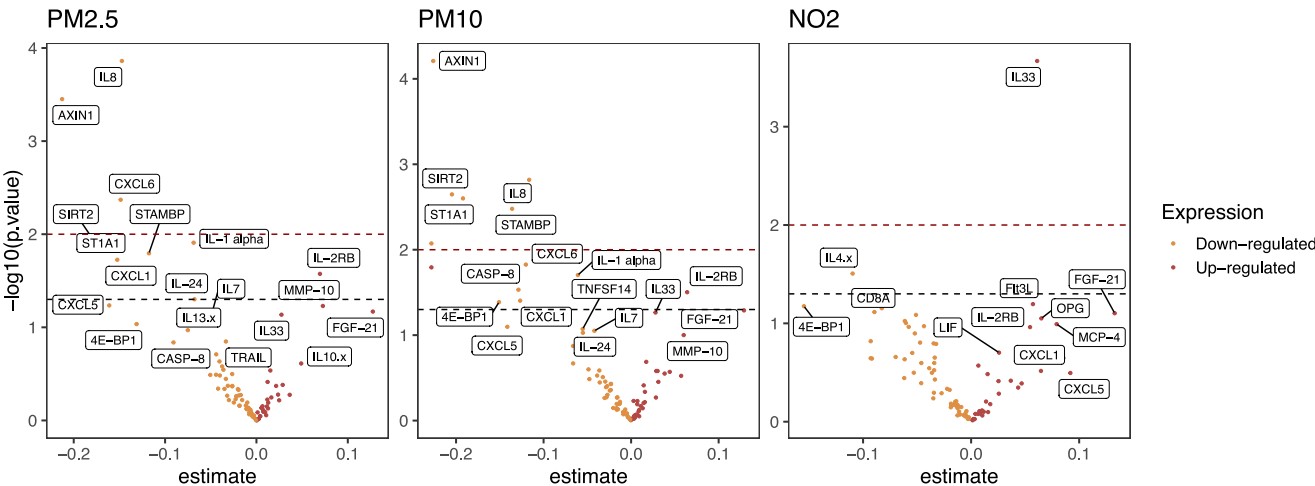

**Fig. 2 | Volcano plots.** Volcano plots of the associations between PM$_{2.5}$, PM$_{10}$, and NO$_2$ and maternal inflammatory proteomic profile in the COPSAC$_{2010}$ cohort from linear regression models. Blue dotted line indicating $p < 0.05$ and red dotted line indicating $p < 0.001$. Source data are provided as a Source Data file.

**Fig. 3 | PCA score and loading plots. A** PCA loading plot showing the distribution of maternal inflammatory proteins at pregnancy week 24 from a principal component analysis in the COPSAC$_{2010}$ cohort. **B** PCA score plots of the maternal inflammatory proteomic profile at pregnancy week 24 showing patterns of correlation from linear regression models between proteins and the distribution of mothers with high vs low (median split) ambient air pollution exposure for PM$_{2.5}$, PM$_{10}$, and NO$_2$.

$PM_{10}$ as illustrated by the Volcano plots in Fig. 2. When analyzing the PC2 score against risk of infections, we found this to be associated with a borderline decreased risk of the overall infection burden; (aIRR: 0.98, 95% CI, 0.97–1.00, $p = 0.052$), which makes sense considering that air pollution downregulated these proteins, but not the different infection subtypes (Supplementary Fig 5).

### Maternal air pollution proteomic fingerprints and risk of early childhood infections in COPSAC$_{2010}$

To explore the associations between prenatal air pollution exposure and the maternal week 24 inflammatory proteomic profile and how these were related to the risk of infections, we used supervised multivariate analysis, sPLS, to generate proteomic fingerprints of air pollution for both $PM_{2.5}$, $PM_{10}$ and $NO_2$ (Fig. 4). We found that a component of 28 proteins for $PM_{2.5}$, 1 protein for $PM_{10}$ (AXIN1) and 19 proteins for $NO_2$ yielded the highest AUC (Fig. 4A–C) and that the scores from the final model (air pollution proteomic fingerprint) were strongly associated with air pollution (Fig. 4D–F). A correlation (Spearman) network analysis of the proteins from each air pollution component is presented in Fig. 5, showing the relationship between each protein according to their classifications. All three fingerprints were highly correlated with each other.

In COPSAC$_{2010}$, the prenatal air pollution proteomic fingerprint for $PM_{2.5}$ was related to a higher risk of total number of infection episodes, cold episodes, pneumonia episodes and fever episodes until age 0-3 years in childhood (Fig. 6). For $PM_{10}$, we found associations with increased risk of total number of infections, colds, pneumonia, tonsillitis, gastric infections and fever episodes and for $NO_2$, associations with total number of infections, colds and gastric infections were demonstrated in Fig. 6.

### Early childhood air pollution proteomic fingerprints and risk of early childhood respiratory infections in EMIL

In the Swedish EMIL cohort ($n = 101$), we computed a comparative air pollution risk score based on the similar proteomic inflammatory panel from Olink, containing the same 92 proteins as in COPSAC, by using the same sPLS model. We then analyzed the associations with the risk of developing respiratory infections risk. Similar to the findings in COPSAC$_{2010}$, the fingerprints, reflecting a high ambient air pollution exposure during the first year of life in the EMIL cohort, associated with the overall respiratory infection risk; $PM_{2.5}$ odds ratio (OR) (95% CI): 1.77 (1.1–3.13), $p = 0.03$, $PM_{10}$: 2.64 (1.23–6.89), $p = 0.02$ and $NO_2$: 1.95 (1.14–3.88), $p = 0.03$ (Fig. 7). Similar to the findings in the COPSAC$_{2010}$ cohort, there was no direct association between air pollution exposure during the first year of life and respiratory infection risk later in early childhood age 1–2 years (Supplementary Table 3).

### AXIN1 levels in pregnancy and early childhood vs risk of early childhood infections and asthma in COPSAC$_{2010}$ and EMIL

The AXIN1 protein was the most dominant protein across all three components of $PM_{2.5}$, $PM_{10}$, and $NO2_2$, with negative loadings in all components, and seemed to play an important role in all fingerprints. Hence, we analyzed this single protein against common infection risk and found that increased levels were associated with a protective role against total number of infections, cold, pneumonia, tonsillitis and fever episodes ($p < 0.05$) (Fig. 8). Interestingly, levels of AXIN1 associated with a decreased risk of asthma risk until age 10 years, adjusted odds ratio (aOR): 0.81 (0.66–0.97), $p = 0.029$ in COPSAC$_{2010}$.

Similar to COPSAC$_{2010}$, AXIN1 levels were associated with a protective effect on risk of respiratory infections, OR (95% CI): 0.50 (0.26–0.98), $p = 0.045$ and aOR: 0.46 (0.21–1.03), $p = 0.059$ in the EMIL cohort, which did not have information on asthma status. These similar findings of both pregnancy and childhood levels suggest a protective effect of the AXIN1 protein levels against risk of early childhood respiratory infections.

## Discussion

This mechanistic study of two independent European birth cohorts, the COPSAC$_{2010}$ and the EMIL cohorts with detailed ambient air pollution exposure assessment of $PM_{2.5}$, $PM_{10}$ and $NO_2$ throughout pregnancy and infancy and with a similar set of inflammatory proteins between the cohorts, demonstrated strong associations between an air pollution proteomic fingerprints and the subsequent risk of developing both upper and lower respiratory tract infections as well as other common infections in early childhood. Further, significant alterations in the maternal inflammation-related proteomic profile from prenatal air pollution were characterized by downregulation of several proteins, including AXIN1, which was also associated with a protective effect on risk of infection and asthma development.

We have investigated the impact of pregnancy and early childhood air pollution exposure on the inflammatory blood proteomic profile in relation to risk of respiratory infections in early childhood. Previously, a large cohort study[12] from MoBA ($n = 17,533$) investigated the association between $NO_2$ exposure in pregnancy and LRTIs until age 18 months in childhood and found no associations. However, they did not assess $PM_{2.5}$ and $PM_{10}$ exposures, concentrations were characterized as low, and diagnoses were based on questionnaires retrospectively at age 6 and 18 months without assessment of daily symptom load as in our study. We found an increased risk of gastric infections from increased $PM_{10}$ exposure, although there were no associations with upper or lower respiratory tract infections directly from air pollution exposure in either the COPSAC$_{2010}$ or the EMIL cohort. However, other larger cohort studies[10,11,18–20] ($n = 1510$, $n = 2568$, $n = 3515$, $n = 2199$ and $n = 1263$, respectively) have previously identified both pre- and postnatal air pollution exposure as an independent risk factor for LRTI development in young children, which could suggest that our study sample size for detecting these direct associations was too small. The ESCAPE project, utilizing data from 10 birth cohorts (BAMSE, GASPII, GINIplus, LISAplus, MAAS, PIAMA, and four INMA cohorts) ($n = 16,059$), found that ambient air pollution exposure from birth increased the risk of pneumonia until age 2 years[13]. A recent (2024) large observational study ($n = 224,214$) found ambient $PM_{2.5}$ exposure in early childhood to be associated with increased risk of respiratory infections in pre-school aged children as well[14].

The mechanisms of disease development have mostly been speculative, including suggestions of a skewed immune system development and negative changes to the fetal and newborn lung development, and we are the first to demonstrate changes in both the maternal and newborn child blood inflammatory proteomic profiles, which were directly related to childhood infection proneness. Current literature describes air pollutants in general to be linked with low-grade systemic inflammation, an altered gut microbiota, and oxidative stress, resulting in an increased risk of cardiovascular disease development in particular[21], but also leading to pathophysiological changes in the respiratory system[12] and increased risk of gastrointestinal disorders[22]. By using mechanistic data layers to understand previously reported associations in other large cohorts between air pollution and respiratory infections, we are now able to characterize a high air pollution inflammatory proteomic profile based on blood plasma samples. We found a downregulation of IL8, confirming the findings of a previously conducted study[15]. Interestingly, the protein AXIN1 seemed to play a vital role in our study as it was significantly downregulated from both $PM_{2.5}$ and $PM_{10}$ exposure and contributed alone to the $PM_{10}$ fingerprint, yielding the highest AUC, and to the $PM_{2.5}$ and $NO_2$ fingerprints with negative loadings as well. Interestingly, AXIN1 has been demonstrated to be highly expressed in the nasopharynx, bronchus, and lung tissue[23], and an experimental mouse study demonstrated reduction of AXIN1 levels in the lungs at early stages of influenza pneumonia. Further, they found AXIN1 to inhibit both RSV and influenza virus replication as well as boost interferon (IFN) response through increased mRNA expression of IFNβ1 and the IFN-targeted

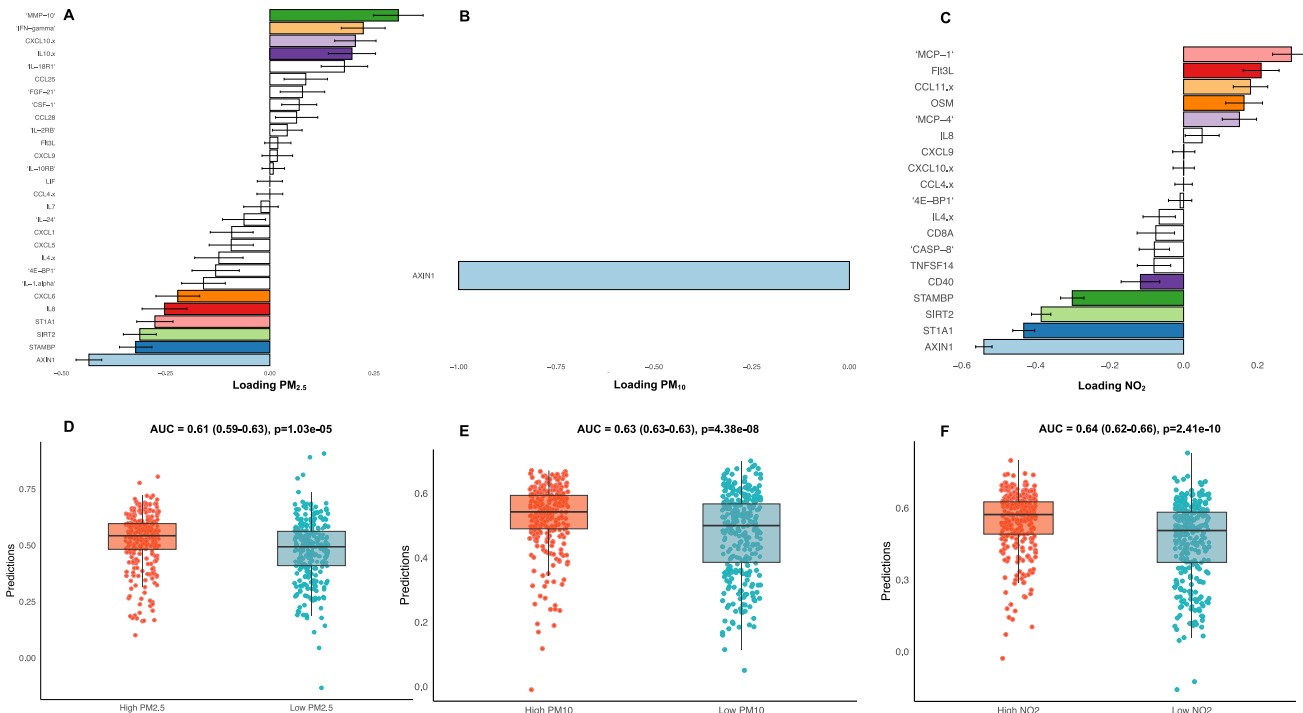

**Fig. 4 | Loading plots of the proteins.** Sparse least squares model predicting high vs low air pollution exposure during pregnancy from maternal pregnancy week 24 proteomic profiles. **A–C** Overview of proteins with corresponding loadings. $PM_{10}$ only had one selected protein (AXIN1). The panels display linear regression estimates with 95% confidence intervals for proteins positively or negatively associated with air pollution measurements ($n = 613$). **D–F** The cross-validated predictions with AUC and p-values, which were strongly associated with air pollution. The box plots represent the class predictions for the median cross-validated model ($n = 613$). The center of the boxes represents the median, their bounds represent the 25th and 75th percentiles, and the lower and upper ends of whiskers represent the smallest and largest values.

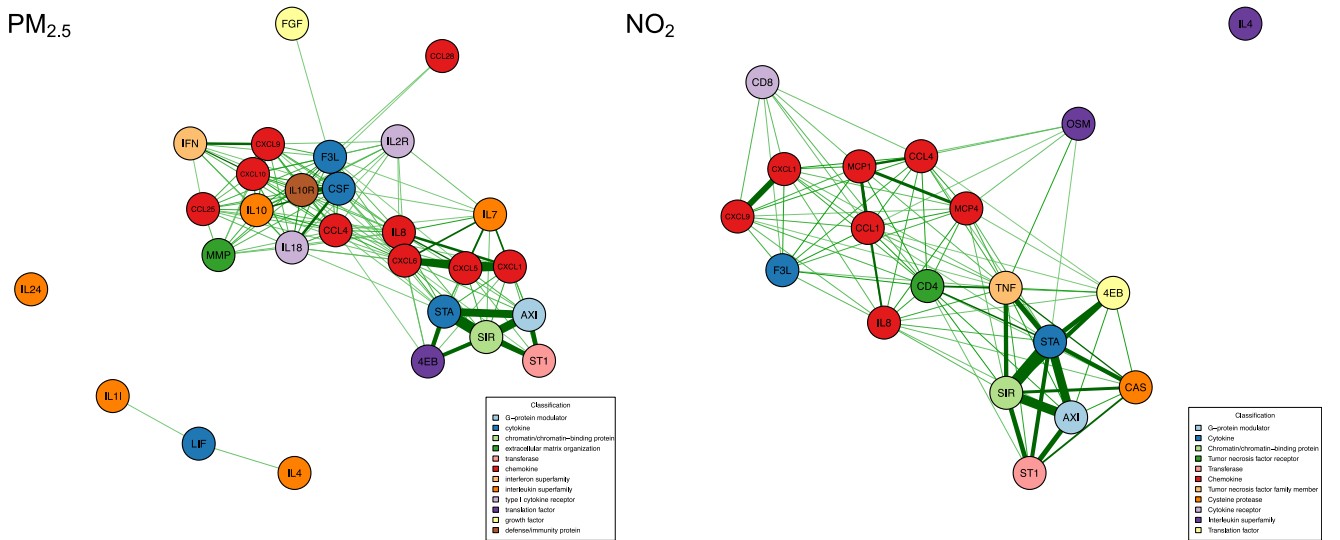

**Fig. 5 | Network analysis.** A Correlation (Spearman) network analysis of the proteins from each air pollution component ($PM_{2.5}$ and $NO_2$). The component for $PM_{10}$ only included 1 protein – AXIN1. Abbreviations: AXI AXIN1, ST1 ST1A1, SIR SIRT2 STA STAMBP, F3L Flt3L, CCL1 CCL11, CD4 CD40, TNF TNFSF14, CAS CASP8, CD8 CD8A, 4EB 4EBP1, MMP MMP10, IFN IFNgamma, IL18R IL18R1, IL1l IL1alpha, FGF FGF21, CSF CSF1, IL2R IL2RB, IL10 IL10RB. Correlations with an absolute value below 0.3 are not shown. Correlations ≥ 0.6 are plotted with maximum edge thickness of the lines. Correlations between 0.3 and 0.6 are scaled proportionally with thinner lines.

anti-viral gene *OAS1* in mice[24]. The authors even suggested the protein as a therapeutic target to prevent airway infections.

Another experimental study of mice recently demonstrated that AXIN1 acted as a regulator of antiviral innate immunity against virus infections by stabilizing interferon regulatory factor 3 (IRF3) and boosting IFN production, also suggesting this protein as an effective antiviral agent[25].

This is in line with the findings from our study suggesting AXIN1 to play a key protective role against risk of infections in young children as we found higher levels of the protein to associate with a decreased risk of the overall infection burden, cold, pneumonia, tonsillitis and fever episodes and asthma diagnosis until age 10 years. Interestingly, similar results showing a decreased risk of respiratory infections from higher AXIN1 levels at age 1 year in the

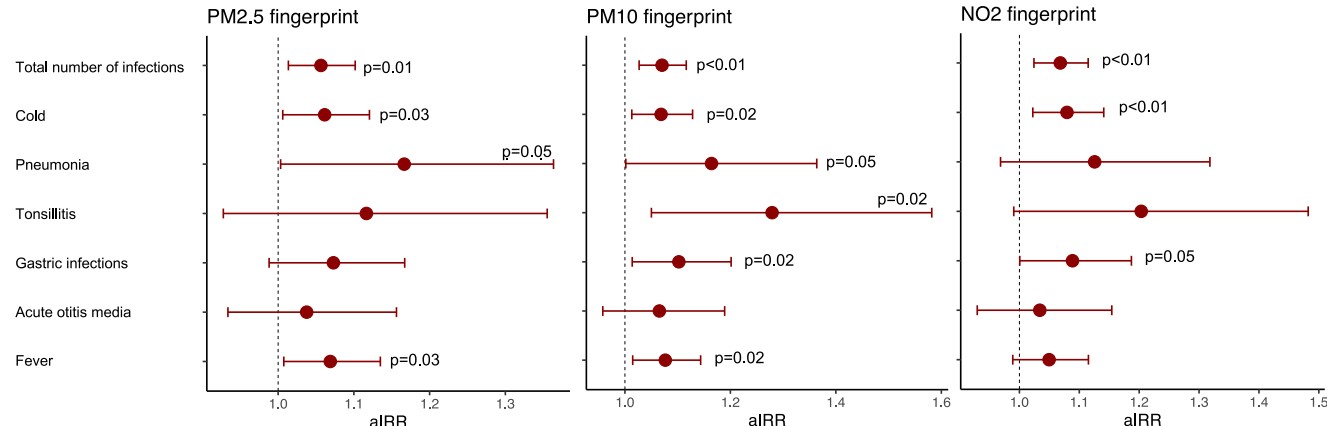

**Fig. 6 | Maternal air pollution proteome vs risk of infections in COPSAC2010.** Association between maternal air pollution proteomic fingerprints from sparse least squares model and risk of infections types age 0–3 years in the COPSAC$_{2010}$ cohort ($n$ = 613). Estimates from Quasi-Poisson regression models with 95% confidence intervals adjusted for gestational age, furred pets during the first year, maternal education and income, time to daycare start, number of older siblings, alcohol use, antibiotic use, and smoking during pregnancy, delivery mode, child hospitalization at birth, and birth season. Source data are provided as a Source Data file.

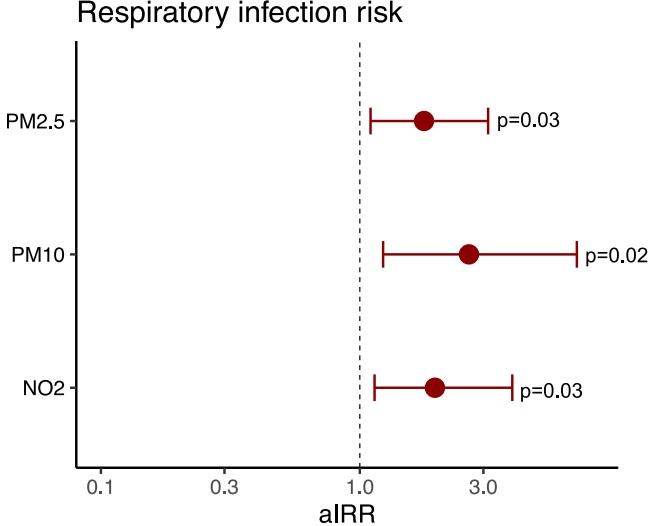

**Fig. 7 | Child air pollution proteome vs risk of infections in EMIL.** Results from the EMIL cohort ($n$ = 101) showing associations between child air pollution proteomic fingerprint and risk of respiratory infections age 1–2 years. Estimates from logistic regression models with 95% confidence intervals adjusted for gestational age, furred pets during the first year, maternal education and income, time to daycare start, number of older siblings, alcohol use, antibiotic use, and smoking during pregnancy, delivery mode, child hospitalization at birth, and birth season. Source data are provided as a Source Data file.

EMIL cohort were demonstrated, supporting the findings from COPSAC$_{2010}$.

In addition, the protein seemed to be directly modified by air pollution exposure, with a significant downregulation in our cohort, highlighting this as a crucial protein in the link between air pollution exposure and increased infection risk found in other studies. These findings in two childhood cohorts indicate what others have previously speculated from experimental studies in animals, that AXIN1 could serve as a potential therapeutic target in the prevention of respiratory infectious diseases, which should be further tested in experimental studies using human cells.

This study was strengthened by the findings from two independent cohorts showing similar associations between the air pollution-derived proteomic fingerprint against mainly respiratory infection risk.

This procedure allows for validation of the initial findings in COPSAC$_{2010}$ where prenatal air pollution caused alterations in the maternal proteome relating to childhood infection risk, by later demonstrating a similar pattern in infancy of the air pollution-derived proteomic fingerprint against later infection risk in EMIL. For the COPSAC$_{2010}$ cohort, the close longitudinal follow-up and unique daily parental registration of infection episodes from birth until age 3 years in combination with a comprehensive amount of information on environmental and social circumstances that could potentially act as confounders and are adjusted for, is a major strength of the study. Together with the proteomic mechanistic data layer, this allows for deep clinical phenotyping of the children and interpretation of the independent impact of air pollution on the mother's proteomic profile that affects the newborn child's infection risk. Further, our infection data in COPSAC$_{2010}$ with daily registrations of symptoms of common infections during the first 3 years of life are unique and different from other cohorts. Another strength is the individual air pollution exposure assessment with high temporal and spatial resolution that takes into account full address history in both cohorts and assessment of ambient air pollution during both pregnancy and infancy whereas most previous large studies have focused on exposure during childhood alone and relied on modeling methods with less fine temporal resolution that did not consider meteorological factors and only had a single address available. The proteomic profiling of the same set of 92 inflammatory proteins in both cohorts followed the same procedure through the Olink Proteomics platform, mitigating the risk of systematic analyses errors. Finally, both cohorts were population-based, including both Danish pregnant women and children, and Swedish children from the greater Copenhagen and Stockholm areas, allowing for generalization of the findings.

Although the similar findings in an independent cohort strengthens our findings, the study was limited by the observational study design with the risk of residual confounding from other potentially associated factors e.g., noise pollution, co-pollutant exposures and lifestyle factors that could influence what the fingerprints also reflects other than air pollution and the overall risk of infections, and even though we adjusted for all available potential confounders from our data we cannot draw causal conclusions from this study. Another limitation is that air pollution assessment was not assessing specific periods of pregnancy. Our study sample size was limited compared with other observational studies linking air pollution to risk of respiratory infections and the lack of overall association between air pollution exposure in pregnancy and childhood infection risk may be

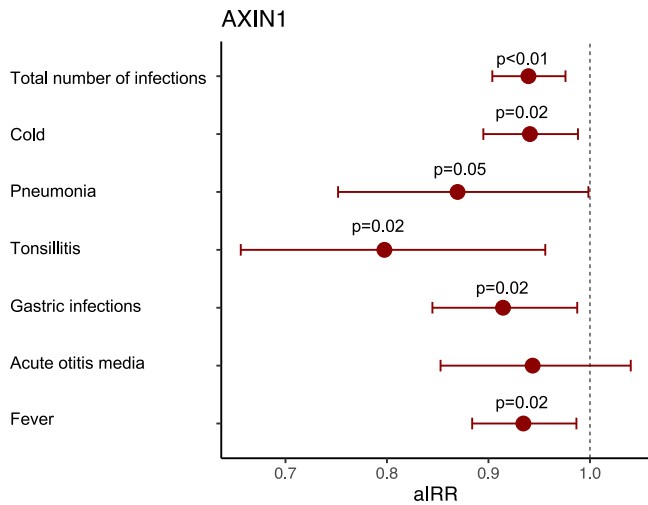

**Fig. 8 | AXIN1 protein levels vs risk of infections in COPSAC2010.** Association between AXIN1 levels and risk of common childhood infections in COPSAC2010 ($n = 613$). Estimates from Quasi-Poisson regression models with 95% confidence intervals adjusted for gestational age, furred pets during the first year, maternal education and income, time to daycare start, number of older siblings, alcohol use, antibiotic use, and smoking during pregnancy, delivery mode, child hospitalization at birth, and birth season. Source data are provided as a Source Data file.

explained by our limited sample size, however, we still had enough participants to demonstrate mechanistic alterations in the blood proteomic profile in relation to infection risk, which was the aim of this study. Hence, we believe this study adds important knowledge to existing literature, trying to understand the underlying mechanisms of previously found associations in larger cohort studies. Finally, the two cohorts were not directly comparable in terms of data availability and should be viewed as complementary, where the COPSAC study investigates prenatal air pollution exposure and the EMIL study investigates infancy air pollution exposure.

We have demonstrated changes in the maternal and newborn child's inflammatory proteomic profile from ambient air pollution exposure relating to both respiratory and common infection risk in early childhood based on data from two independent European birth cohorts. In addition, our findings suggest that the AXIN1 protein, which was modified by air pollution exposure to play a role in the downregulation of infection proneness in young children and could potentially serve as a target protein in respiratory infection prevention.

## Methods

### The COPSAC$_{2010}$ cohort
The study population and baseline characteristics of the Danish Copenhagen Prospective Studies on Asthma in Childhood 2010 (COPSAC$_{2010}$) cohort have been described in detail previously[26,27]. In brief, 700 mother-child pairs were followed from pregnancy week 24 and throughout early childhood with a longitudinal deep clinical phenotyping in the COPSAC clinic, including daily parental infection diaries during the first 3 years of life. The study was approved by the local ethics committee (H-B-2008-093). Parents gave written informed consent before enrollment.

### Longitudinal clinical follow-up
Children were followed in the clinic from week 1 until age 3 years; at age 1 week, 1 month, 3 months, 6 months, 1 year, 18 months, 2 years, 30 months and 3 years where symptom diaries were assessed at each visit in addition to acute visits when experiencing any respiratory symptoms for doctor diagnoses. At every visit, children were

diagnosed, and medications were registered continuously throughout this period.

### Daily registered infection episodes from parental diaries
Daily diary cards were filled out by parents during the first 3 years of life with registrations of any symptoms of cold (upper respiratory tract infection symptoms), gastrointestinal infections (acute diarrhea or vomiting symptoms), any fever (>38 °C) and doctor diagnosed acute otitis media, tonsillitis and pneumonia as previously detailed[17,28]. Diaries and diagnoses were reviewed with the families by the COPSAC physicians during the continuous, planned clinical visits to validate the symptom and disease entries. All episodes of infections lasting at least three consecutive days were entered into the database and double-checked[17].

### Ambient air pollution assessment
The unique personal identification numbers from the Danish civil registration system allowed us to link the COPSAC data with individual maternal home addresses throughout pregnancy from conception until birth, including registrations of moving, emigration, immigration, and death[29]. Concentrations of $PM_{2.5}$, $PM_{10}$, and $NO_2$ were calculated from each home address using the Danish Eulerian Hemispheric Model (DEHM)-urban background model (UBM)-Danish Air Pollution and Human Exposure Modeling System (AirGIS) model system[30–32]. For each participant, we calculated time-weighted mean from conception to birth, reflecting mean exposure during pregnancy. We fitted models with increments for each exposure, separately, corresponding to the interquartile range (IQR) for the full study population. Further descriptions of the air pollution data collection and processing have been described in detail in our previous publication[2].

### Proteomic inflammatory data
Plasma proteins were assessed using the Olink Target 96 Inflammation panels from Olink (Uppsala, Sweden). Blood plasma samples were thawed, randomized across nine 96-well plates, refrozen, and transferred to Olink on dry ice before being screened with the Olink Target 96 Inflammation assay[33]. A total of 92 inflammatory proteins were quantified from our samples. The results were presented as log2-transformed Normalized Protein Expression (NPX) values. All plates and samples passed quality control (QC) thresholds, defined as an internal control SD of <0.2 NPX from the median between plates and <0.3 NPX between samples within plates, and the samples with QC warnings were excluded from the analyses. The complete list of all 92 inflammatory proteins, including the limit of detection (LOD) and percentage below LOD, is detailed in Supplementary Table 1 with associations visualized in a heatmap in Supplementary Fig 1.

### Covariates
We previously demonstrated lifestyle differences between children who lived in an urban versus rural environment (see previous publication for details[28]), which are strongly correlated to air pollution exposure, including exposure to furred pets during the first year, maternal education and income, time to daycare start, and number of older siblings[28]. In addition, we considered gestational age, alcohol use, antibiotic use, and smoking during pregnancy, delivery mode, child hospitalization at birth, and birth season could influence our results and therefore included these in our analyses as potential confounders[28].

### The EMIL cohort
The study population and baseline characteristics of the Swedish birth cohort Etiological Mechanism of air pollution effects on the Infant Lungs (EMIL) have previously been described as well[15]. In brief, parents of children born from 2014-2017 in Stockholm, Sweden, were invited to participate with the purpose of investigating biological mechanisms

behind the adverse respiratory health effects of air pollution in children during early life[15]. At age 3 months, parents were interviewed regarding social and environmental characteristics. Ambient air pollution concentrations were calculated from birth until age 1 year, and time-weighted annual means of $PM_{2.5}$, $PM_{10}$, and $NO_2$ were calculated based on residential history and estimated annual levels adjusted for short-term variations using urban background monitor measurements[15]. The children underwent blood sampling at age 1 year ($n = 101$) for proteomic analyses of inflammation-related proteins measured by Olink, which was the exact same inflammatory panel, including 92 proteins, as in COPSAC$_{2010}$. Airway infection assessment from age 1 to 2 years was a combined variable including croup, pneumonia, or bronchitis, assessed by parental interviews at age 2 years. The study was approved by the Swedish ethics committee from the Regional Ethics Review Board, Karolinska Institute, Stockholm, Sweden. All caregivers provided written informed consent.

### Statistical analyses

In our analysis, we included COPSAC$_{2010}$ children with more than 30 days of diary registrations in total[34]. We analyzed the risk of infection episodes aged 0–3 years from prenatal ambient air pollution exposure in a Quasi-Poisson regression model, estimating the incidence rate ratio (IRR). This statistical method was used to account for overdispersion in the data[35]. The associations between prenatal ambient air pollution and each protein in the inflammatory panel were analyzed using linear regression and are illustrated for each exposure by Volcano plots, where we included all proteins independent of LOD as recommended by Olink and similar to previous studies[36]. We performed a principal component analysis (PCA) of the 92 maternal inflammatory proteins and investigated the scores in relation to air pollution exposure and risk of infections. We then aimed to identify the set of proteins during pregnancy most strongly associated with $PM_{2.5}$, $PM_{10}$, and $NO_2$ to derive an air pollution proteome risk score (fingerprint) for each mother. This was achieved using a supervised sparse partial least square (sPLS) model selecting the optimal set of proteins through 10-fold cross-validation repeated 10 times based on the Area Under the Curve (AUC) as we previously did[37]. The final sPLS model was chosen for its highest median AUC value and lowest variance across cross-validation rounds. A predicted pollution-risk score was calculated for each individual by multiplying the original data matrix by the loadings derived from the final model, which were subsequently associated with offspring infection episodes. We used the same model based on air pollution in COPSAC$_{2010}$ to derive a similar proteomic-pollution risk score in the EMIL cohort in childhood and associated the predicted scores against infections in early childhood in both cohorts, an approach used in our previous study[38]. Finally, we identified the most contributing protein of the sPLS fingerprints and associated this against the infection outcomes as well as asthma (previously defined in refs. 26,28) until age 10 years in COPSAC$_{2010}$ in a Quasi-Poisson and logistic regression model, respectively. All analyses were performed with R software (v4.0.3; R Foundation for Statistical Computing, Vienna, Austria). $P$ values <0.05 were considered statistically significant.

### Reporting summary

Further information on research design is available in the Nature Portfolio Reporting Summary linked to this article.

## Data availability

For the COPSAC$_{2010}$ cohort dataset, participant-level personally identifiable data are protected under the Danish Data Protection Act and European Regulation 2016/679 of the European Parliament and of the Council that prohibit distribution even in pseudo-anonymised form. However, participant-level data can be made available under a data-transfer agreement as part of a collaboration effort with COPSAC. For EMIL–participant-level personally identifiable data are protected under the Swedish Data Protection Act (2018:218) and European Regulation 2016/679 of the European Parliament and of the Council. Participant-level data can be made available under a data transfer agreement as part of a collaboration with COPSAC (nicklas.brustad@dbac.dk) or EMIL (olena.gruzieva@ki.se). Source data are provided with this paper.

## Code availability

Analytical code will be available upon request to the corresponding author with publication to researchers whose proposed use of the data and code has been approved.

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

## Acknowledgements

We express our deepest gratitude to the children and families of the COPSAC_{2010} cohort study for all their support and commitment. We acknowledge and appreciate the unique efforts of the COPSAC research team.

## Author contributions

NB has written the first draft of the manuscript. NB and TW performed the statistical analyses. S.H., C.T.P., O.G., L.C., G.P., M.A., J.K., M.P., M.K., H.H., D.A., E.M., J.T., J.S., K.B., and B.C. have provided important intellectual input and contributed considerably to the interpretation of the data. The corresponding author had full access to the data and had final responsibility for the decision to submit for publication.

## Competing interests

The authors declare no competing interests.

## Additional information

**Nicklas Brustad** [1] ✉, **Tingting Wang** [1], **Shizhen He**[2], **Casper-Emil Tingskov Pedersen** [1], **Olena Gruzieva** [2,3], **Liang Chen** [1], **Göran Pershagen**[2], **Mina Ali**[1], **Julie Nyholm Kyvsgaard**[1], **Marie Pedersen**[4], **Matthias Ketzel**[5], **Heikki Hyöty** [6], **Daniel Agardh** [7], **Erik Melén** [8], **Jonathan Thorsen** [1,9], **Jakob Stokholm** [1,10,11], **Klaus Bønnelykke** [1,9] & **Bo Chawes** [1,9]

[1]COPSAC, Copenhagen Prospective Studies on Asthma in Childhood, Copenhagen University Hospital - Herlev and Gentofte, Copenhagen, Denmark. [2]Institute of Environmental Medicine, Karolinska Institutet, Stockholm, Sweden. [3]Centre for Occupational and Environmental Medicine, Region Stockholm, Stockholm, Sweden. [4]Department of Public Health, University of Copenhagen, Copenhagen, Denmark. [5]Department of Environmental Science, Aarhus University, Roskilde, Denmark. [6]Faculty of Medicine and Health Technology, Tampere University, and Fimlab Laboratories, Tampere, Finland. [7]Department of Clinical Sciences, Lund University, Malmö, Sweden. [8]Department of Clinical Science and Education, Södersjukhuset, Karolinska Institutet, Stockholm, Sweden. [9]Department of Clinical Medicine, University of Copenhagen, Copenhagen, Denmark. [10]Department of Pediatrics, Slagelse Sygehus, Næstved, Denmark. [11]Department of Food Science, University of Copenhagen, Copenhagen, Denmark. ✉e-mail: nicklas.brustad@dbac.dk

