## [Transparent Peer Review file · Nature Communications]

Air pollution-induced proteomic alterations increase the risk of child respiratory infections

Corresponding Author: Dr Nicklas Brustad

Version 0:

Reviewer comments:

Reviewer #1

(Remarks to the Author)

This is an observational epidemiological study investigating prenatal and postnatal exposure to air pollutants in relation to changes to proteomics in mothers/children, and the risk of respiratory infection. The study identified that across the three studied pollutants, PM_{2.5}, PM₁₀ and NO₂, AXAN1 was downregulated by higher exposure to PM, and that may have implications on respiratory infection. This is a novel finding particularly in human participants in this research field. However, there are some limitations which means cautious interpretation is needed. Some details were also lacking in this manuscript. I have a few comments.

Title: I found the title was a bit unnecessarily long.

Abstract: The abstract needs a lot of improvement for readability and clarity.

N=714 should clearly show the exact study number for each cohort.

Line 56-60, as far as I read it, this is not the MAIN aim of this study, it is the proteomics pathways between air pollution and respiratory infection to be investigated. Please revise accordingly.

Little mention on the statistical methods used.

Interpretation of results (line 69-70) was confusing as it reads 'downregulated AX1N1...significantly decrease...', this seems to suggest that air pollution indirectly decrease the infection risk, and also in contrast with line 278-279. If I interpreted it right, it should be "air pollution-induced downregulated AX1N1...significantly INCREASE the risk of respiratory infection"? I would suggest to use the same tone in expressing the results, as it currently read that air-pollution induced proteomics changes were associated with a higher risk of respiratory infection, and then later AX1N1 has a protective effect...

Introduction: perhaps useful to elaborate a bit the study findings in reference 15 – what inflammation proteins have been found to be associated with air pollution? And are there any non air pollution-related studies have linked these proteins to respiratory infection risk?

Methods:

first of all, I am not entirely sure whether COSPAC2010 could be named as discovery data, and then EMIL as replication. Blood sample had been collected in different times, and pregnancy is such a specific period, how these two cohorts could be comparable? I did not see a strong justification? It would have made much more sense if blood sample also collected at age 1 year in the COSPAC2010 cohort?

Line 121- what kind of 'clinical measurements'? Line 123 'several social and environmental factors' what are these? Were they time-varying, and considered in your statistical models?

Daily infection data in the COSPAC cohort was basically self-reported over 3 years. Although it was stated that these are validated by physicians (line 131), I was wondering how? Clinical visits were only scheduled every few months, and how the daily symptoms in the past few months could be validated? The decision of only including episodes lasting three days should also be justified, perhaps to cite clinical guidelines?

Line 134-144: I am not entirely sure whether daily mean of air pollutants were computed for each participants? If only average mean concentration during the entire pregnancy was used in the analyses? If so, need to state that.

In terms of statistical analyses – some details were not mentioned, for example, PCA was mentioned later in the Results, but was not explained here in this section. Please double check and make sure all the methods you have used were clearly explained here. I was also wondering why additional analyses were not done in the EMIL cohort regarding air pollution during pregnancy and proteomics at aged 1 year, and infection in 1-2 years? At least this could complement the COSPAC analyses.

Results: seemed OK, although number of figures could be reduced?

Discussion: largely fine, but in terms of limitations, co-pollutant exposures (or exposure mixtures) were not mentioned given

that it is not possible to investigate in this study as they were highly correlated. Sensitive time-window of pregnancy and postnatal periods have not been investigated. And as I said before, the analyses between the two cohorts perhaps are not directly comparable?

Reviewer #2

(Remarks to the Author)

The study by Brustad et al. investigated the impact of early exposure to air pollution on respiratory infection in early childhood in two cohorts integrating proteomic analysis to understand if a specific proteomic profile is associated with future risk for respiratory infections. The manuscript assess a relevant knowledge gap, that is to better understand the mechanistic link between air pollution exposure and respiratory symptoms.

I have several comments on how to improve the manuscript.

Major comment: The authors state that the protein AXIN1 has a protective effect for subsequent respiratory tract infections. In the abstract, you state that higher air pollution decreases ASIN1 and that this downregulation decreases the risk for respiratory tract infections (written in the abstract). This, however contradicts the statement in the manuscript (lines 275-286) where "increased levels were associated with a protective role? Please excuse my ignorance, but this is really contradictory. Also, what is the effect of air pollution then? Higher air pollution decreases its levels, which is good, then air pollution has a protective effect for resp. symptoms? This is counter intuitive to me.

Abstract: Please include relevant information on the cohort e.g. number of participants. The information on p-values is not informative, but actual number of participants, events, and coefficients of associations should be reported. I suggest to adhere to strobe guidelines on how to report your data.

Introduction: I hypothesize is missing, although stated in the graphical abstract. Please be clear if there is a hypothesis or if this is an explorative study.

Methods (lines 135-144): Have the authors considered the place of work of the mother?

Lines (147-153): When were the blood samples of the mother exactly taken? I am asking this question since maybe it could be relevant to assess the air pollution more specifically at that time, rather than taking only the prenatal mean of exposure? Other this have shown that there maybe more sensitive periods of lung development in relation to air pollution in different trimesters.

Because of the two different cohort settings, the samples were taken once from the mother, and then from the child. How comparable is the proteomic data when samples were taken from these different sources?

Results (lines 215-218) What is the mechanistic link between GI infections and air pollution? This interesting finding requires more elaboration, I think.

Statistics: I am not an expert on PCA Analysis, maybe some other experts should be involved. Lines 230-232: PC2 Loading explains 6.5 % of the variation. Is this good loading, meaning to explain a lot of the findings, or is this association rather weak maybe? I would appreciate an explanation.

Reviewer #3

(Remarks to the Author)

Review of manuscript NCOMMS-25-14157. This manuscript assessed data from two cohort studies using proteomics from mother/children and assessed the associations of the proteomics linking air pollution and childhood infections. The advantages of this study is that this study has relatively large sample size and has both testing and validation cohorts. However, I have a few questions and concerns on some methods and results interpretation of this manuscript before I can recommend accepting of this manuscript. Thanks.

Major Comments:

1. In line 183, authors used Quasi-Poisson regression model to estimate the incidence rate ratio with air pollution. Do this mean the outcome is the count of each infection for each child during 0-3 years? This analysis method will not use the longitudinal nature of the data, I wonder whether authors use this statistical method, instead of using generalized linear mixed model or time-to-event analysis? Another limitation of this model is that the maximum count of the infection could be bounded by visit time, which may violate the model assumption. In addition, looking at Fig 1, the infection categories changed dramatically as children age increased, using a sum count of infection will loss this information.
2. In lines 205, authors mentioned 614 (88%) children had available diary data, which are the main missing factors contributing to the missing data? Authors should provide a flow chart in the Supplement showing how the participants were removed due to missingness.
3. For the analysis between air pollution and proteomics, did authors only used raw p-value < 0.05 as cut-off? Since there are 92 proteomic markers in the analysis, some adjustment of false discovery is needed for results presented in Fig 2.
4. For Fig 4, I wonder what does that mean for the y-axis (predictions) for Fig D-F?
5. For Fig 5, I wonder why AXIN1 is not shown up in the PM2.5 and NO2 signature even though AXIN1 has large loading in the sPLS regression with PM2.5 and NO2.
6. In lines 266 – 267, authors stated no fingerprints were similar between EMIL and COPSA2010, but I did not see the fingerprints results presented for EMIL study.
7. This study has a testing cohort and a validation cohort, which is a plus compared to several previous studies. However, there are some discrepancies between testing analysis and validation analysis. For example, no associations between air pollution and respiratory infection in COPSAC2010 cohort, which makes the conclusion of the manuscript less convincing. Again, this might be the statistical methods used to assess associations between air pollution and childhood infection.

Minor Comments:

1. The title of the manuscript seems too long. Please shorten it based on journal's requirements.
2. For Fig 5, is the link between each proteomic feature is shown as correlation? If so, please provide a legend for the color and thickness of the link for the correlation.

Version 1:

Reviewer comments:

Reviewer #1

(Remarks to the Author)

Thanks to the authors for addressing my and other reviewers' comments. The manuscript did improve a lot, and I have no further comments.

Reviewer #2

(Remarks to the Author)

Reviewer #3

(Remarks to the Author)

I would like to thank the authors for revising the manuscript and addressing my questions. My comments are adequately addressed. I have a few additional comments and suggestions:

1. For the sPLS-derived proteomic fingerprints of air pollution, I wonder if the prediction scores are presented in Figures 4D-F? If so, authors should make it clear that the prediction is the risk score/fingerprint.
2. I believe the proteomic fingerprint of air pollution is a key analysis in the manuscript; the authors should provide more technical details, such as formulas and equations, or share the sPLS regression and analysis code for deriving the risk scores, thereby eliminating the need for the research community to request these details for better transparency.
3. Although the authors provided explanations for the study's potential lack of power to detect air pollution (especially PM2.5 and NO2) and childhood infections, I still believe that further discussion is needed to explain the absence of associations between PM2.5/NO2 and infections, despite the proteomic fingerprint scores being associated with infections.

Thank you.

REVIEWER COMMENTS

Reviewer #1 (Remarks to the Author):

This is an observational epidemiological study investigating prenatal and postnatal exposure to air pollutants in relation to changes to proteomics in mothers/children, and the risk of respiratory infection. The study identified that across the three studied pollutants, PM_{2.5}, PM₁₀ and NO₂, AXIN1 was downregulated by higher exposure to PM, and that may have implications on respiratory infection. This is a novel finding particularly in human participants in this research field. However, there are some limitations which means cautious interpretation is needed. Some details were also lacking in this manuscript.

I have a few comments.

Title: I found the title was a bit unnecessarily long.

RESPONSE The title has been changed from

"Air pollution-induced maternal and child proteomic alterations increase the risk of childhood respiratory infections identifying AXIN1 protein as a potential target for asthma and respiratory infection prevention: Evidence from two European birth cohorts"

to:

"Air pollution-induced proteomic alterations increase the risk of child respiratory infections identifying AXIN1 as a potential target for prevention"

Abstract: The abstract needs a lot of improvement for readability and clarity.

N=714 should clearly show the exact study number for each cohort.

Line56-60, as far as I read it, this is not the MAIN aim of this study, it is the proteomics pathways between air pollution and respiratory infection to be investigated. Please revise accordingly.

RESPONSE We agree. This has now been added:

*Page 4, lines 56-60: "We utilized data from two independent prospective birth cohorts to investigate the influence of prenatal and postnatal ambient air pollution exposure of PM_{2.5}, PM₁₀ and NO₂ **on maternal and child proteomic profiles and the** risk of daily diary-registered common infections age 0-3 years in the Danish COPSAC₂₀₁₀ (**n=613**) and pneumonia, croup and bronchitis age 1-2 years in the Swedish EMIL (**n=101**)."*

Little mention on the statistical methods used.

RESPONSE This has now been added:

Page 4, lines 63-65: “A supervised sparse partial least square model generated proteomic fingerprints of air pollution $PM_{2.5}$, PM_{10} and NO_2 in both cohorts **that was analyzed against infection outcomes using Quasi-Poisson and logistic regression models, respectively.**”

Interpretation of results (line 69-70) was confusing as it reads ‘downregulated AX1N1...significantly decrease...’, this seems to suggest that air pollution indirectly decrease the infection risk, and also in contrast with line 278-279. If I interpreted it right, it should be “air pollution-induced downregulated AX1N1...significantly INCREASE the risk of respiratory infection”? I would suggest to use the same tone in expressing the results, as it currently read that air-pollution induced proteomics changes were associated with a higher risk of respiratory infection, and then later AX1N1 has a protective effect...

RESPONSE We apologize for the confusion. The sentences in the Abstract have been rephrased to clearly describe that higher levels of AXIN1 (a protein which is downregulated by air pollution) protect against infection and asthma outcomes:

Page 4-5, lines 71-77. “**Higher AXIN1 protein levels** associated with significantly decreased risks of total number of infections, cold, pneumonia, tonsillitis and fever episodes, and asthma risk until age 10 years in COPSAC₂₀₁₀ and a significantly decreased risk of respiratory infections in EMIL ($p < 0.05$) suggesting a protective effect of this specific protein in both cohorts. Finally, **higher levels of AXIN1** associated with a decreased risk of asthma risk until age 10 years, adjusted odds ratio: 0.81 (0.66-0.97), $p = 0.029$ in COPSAC₂₀₁₀.”

Introduction: perhaps useful to elaborate a bit the study findings in reference 15 – what inflammation proteins have been found to be associated with air pollution? And are there any non air pollution-related studies have linked these proteins to respiratory infection risk?

RESPONSE: We agree, and this has now been added to the Introduction. Interestingly, they have similar findings on the downregulation of IL-8 as we find, which we have also added to our Discussion. Thanks for the great suggestion.

Page 6, lines 101-106: “Finally, air pollution exposure in early life have been associated with alterations in the inflammatory proteomic profile **showing e.g. downregulation of IL-8¹⁵. In another study, overexpression of IL-8 in lung epithelium has shown to benefit lung immunity to bacterial infection¹⁶.** However, these mechanistic pathways are yet to be investigated in relation to childhood respiratory infection risk.”

Page 18, lines 341-3345 “By using mechanistic data layers to understand previously reported associations in other large cohorts between air pollution and respiratory infections, we are now able to characterize a high air pollution inflammatory proteomic profile based on blood plasma samples. **We found a downregulation of IL-8 confirming the findings of a previously conducted study¹⁵.**”

Methods:

first of all, I am not entirely sure whether COSPAC2010 could be named as discovery data, and then EMIL as replication. Blood sample had been collected in different times, and pregnancy is such a specific period, how these two cohorts could be comparable? I did not see a strong justification? It would have made much more sense if blood sample also collected at age 1 year in the COSPAC2010 cohort?

RESPONSE: We have deleted the discovery and validation/replication terms throughout the manuscript.

Line 121- what kind of 'clinical measurements'?

Line 123 'several social and environmental factors' what are these? Were they time-varying, and considered in your statistical models?

RESPONSE This clinical measurements refer to measurements that are actually not relevant for this study and this has been removed. Further, the sentence about social and environmental factors have been removed as this might confuse the reader as the covariates adjusted for are detailed in the "Covariates" section in Methods:

Daily infection data in the COSPAC cohort was basically self-reported over 3 years. Although it was stated that these are validated by physicians (line 131), I was wondering how? Clinical visits were only scheduled every few months, and how the daily symptoms in the past few months could be validated? The decision of only including episodes lasting three days should also be justified, perhaps to cite clinical guidelines?

RESPONSE The parents filled out a diary registering every single day with or without symptoms of the children. Hence, when they visited the COPSAC clinic, a physician had a conversation about every registration made in the diary of the child since the last visit to confirm with the parents. The visits are relatively often done with only months interval over a 3-year period. In combination with the parents' daily symptom registrations of the child, no other cohort have more detailed information on early childhood infection episodes. In addition, they visited the clinic if the child had any respiratory symptom as an acute visit, where this would be registered as well by the physicians meaning that the episodes are not only self-reported. We have previously published this in detail in other studies (e.g. Bisgaard, NEJM, 2016, Brustad, JAMA Network Open, 2025, Kyvsgaard, JACI, 2024). We have updated the sentence to include more information:

Page 8, lines 123-128: "Children were followed in the clinic from week 1 until age 3 years; at age 1 week, 1 month, 3 months, 6 months, 1 year, 18 months, 2 years, 30 months and 3 years **where symptom diaries were assessed at each visit in addition to acute visits when experiencing any respiratory symptoms for doctor diagnoses.** At every visit, children were diagnosed, and medication were registered continuously throughout this period."

In addition to the above, acute otitis media, tonsillitis and pneumonia were also registered if the children had visited and got these diagnoses at the hospital or general practitioner.

The reason for including an episode if it lasted at least three days, is a decision we have previously taken and is used when analyzing our infection diary data to capture episodes of a certain length and severity. We have now added references of our previous studies to the sentence.

Line 134-144: I am not entirely sure whether daily mean of air pollutants were computed for each participants? If only average mean concentration during the entire pregnancy was used in the analyses? If so, need to state that.

RESPONSE : This has now been added:

Page 9, lines 146-147: *"For each participant, we calculated time-weighted mean from conception to birth **reflecting mean exposure during pregnancy.**"*

In terms of statistical analyses – some details were not mentioned, for example, PCA was mentioned later in the Results, but was not explained here in this section. Please double check and make sure all the methods you have used were clearly explained here. I was also wondering why additional analyses were not done in the EMIL cohort regarding air pollution during pregnancy and proteomics at aged 1 year, and infection in 1-2 years? At least this could complement the COSPAC analyses.

RESPONSE: We have now added details about the principal component analysis to the statistical analysis part:

Page 11, lines 194-196: ***"We performed a principal component analysis (PCA) of the 92 maternal inflammatory proteins and investigated the scores in relation to air pollution exposure and risk of infections."***

Regarding the suggested pregnancy analyses of The EMIL cohort, there is no data collected on air pollution before birth unfortunately.

Results: seemed OK, although number of figures could be reduced?

RESPONSE: We would like to keep all figures. But we would of course reduce the number of figures if the editors find this necessary for publication.

Discussion: largely fine, but in terms of limitations, co-pollutant exposures (or exposure mixtures) were not mentioned given that it is not possible to investigate in this study as they were highly correlated. Sensitive time-window of pregnancy and postnatal periods have not been investigated. And as I said before, the analyses between the two cohorts perhaps are not directly comparable?

RESPONSE: This has now been added to the discussion:

Page 21, lines 390-395: ***"Although the similar findings in an independent cohort strengthens our findings, the study was limited by the observational study design with the risk of residual confounding from other potentially associated factors e.g. noise pollution, co-pollutant exposures and lifestyle factors that could influence what the fingerprints also***

reflects other than air pollution and the overall risk of infections, and even though we adjusted for all available potential confounders from our data we cannot draw causal conclusions from this study”

and

Page 21, lines 402-405: ***Finally, the two cohorts were not directly comparable in terms of data availability and should be viewed as complementary, where the COPSAC study investigates prenatal air pollution exposure and the EMIL study investigates infancy air pollution exposure.***

We believe our study is somehow investigating the sensitive time-window of pregnancy (COPSAC) and postnatal period (EMIL) as these periods are covered in the air pollution measurement periods. However, we acknowledge that the study could not investigate specific periods during pregnancy which could be more crucial than others and have now mentioned this:

Page 21, lines 397-398: ***“Another limitation is that air pollution assessment was not assessing specific periods of pregnancy.”***

Reviewer #2 (Remarks to the Author):

The study by Brustad et al. investigated the impact of early exposure to air pollution on respiratory infection in early childhood in two cohorts integrating proteomic analysis to understand if a specific proteomic profile is associated with future risk for respiratory infections. The manuscript assess a relevant knowledge gap, that is to better understand the mechanistic link between air pollution exposure and respiratory symptoms. I have several comments on how to improve the manuscript.

RESPONSE: Thank you.

Major comment: The authors state that the protein AXIN1 has a protective effect for subsequent respiratory tract infections. In the abstract, you state that higher air pollution decreases ASIN1 and that this downregulation decreases the risk for respiratory tract infections (written in the abstract). This, however contradicts the statement in the manuscript (lines 275-286) where “increased levels were associated with a protective role? Please excuse my ignorance, but this is really contradictory. Also, what is the effect of air pollution then? Higher air pollution decreases its levels, which is good, then air pollution has a protective effect for resp. symptoms? This is counter intuitive to me.

RESPONSE We apologize for the confusion. The sentences in the Abstract have been rephrased to clearly describe that higher levels of AXIN1 (a protein which is downregulated by air pollution) protect against infection and asthma outcomes:

Page 4-5, lines 71-77. ***“Higher AXIN1 protein levels associated with significantly decreased risks of total number of infections, cold, pneumonia, tonsillitis and fever episodes, and asthma risk until age 10 years in COPSAC₂₀₁₀ and a significantly decreased***

*risk of respiratory infections in EMIL ($p < 0.05$) suggesting a protective effect of this specific protein in both cohorts. Finally, **higher levels of AXIN1** associated with a decreased risk of asthma risk until age 10 years, adjusted odds ratio: 0.81 (0.66-0.97), $p = 0.029$ in COPSAC₂₀₁₀.”*

Abstract: Please include relevant information on the cohort e.g. number of participants. The information on p-values is not informative, but actual number of participants, events, and coefficients of associations should be reported. I suggest to adhere to strobe guidelines on how to report your data.

RESPONSE We have adhered to the formatting guidelines of Nature Communications, which is stating: *The abstract — which should be no more than 200 words long and contain no references — should serve both as a general introduction to the topic and as a brief, non-technical summary of the main results and their implications.* These abstract usually do not include information on number of participants, events etc.

Introduction: I hypothesize is missing, although stated in the graphical abstract. Please be clear if there is a hypothesis or if this is an explorative study.

RESPONSE We have now added our hypothesis to the Introduction:

Page 7, lines 110-112: ***“We hypothesize that air pollution exposure in early life changes the inflammatory proteomic profile of the pregnant mother and newborn child with associations to later respiratory infection risk.”***

Methods (lines 135-144): Have the authors considered the place of work of the mother?

RESPONSE Yes, unfortunately this information was not available.

Lines (147-153): When were the blood samples of the mother exactly taken? I am asking this question since maybe it could be relevant to assess the air pollution more specifically at that time, rather than taking only the prenatal mean of exposure? Other this have shown that there maybe more sensitive periods of lung development in relation to air pollution in different trimesters.

RESPONSE No, unfortunately this information was not available. However, we have now mentioned this as a limitation of the study:

Page 21, lines 397-398: ***“Another limitation is that air pollution assessment was not assessing specific periods of pregnancy.”***

Because of the two different cohort settings, the samples were taken once from the mother, and then from the child. How comparable is the proteomic data when samples were taken from these different sources?

RESPONSE We believe the proteomics data are very comparable given that they are using an identical panel and both analyzed by the same Olink platform on the same scale

providing NPX values, although it is taken from the mother and the child, which we consider a strength as it represents air pollution exposure both in pregnancy and infancy. The analyses of the two cohorts should be considered as complementary and not identical, which we have now stated in the Discussion as well:

Page 21, lines 405-408: ***“Finally, the two cohorts were not directly comparable in terms of data availability and should be viewed as complementary, where the COPSAC study investigates prenatal air pollution exposure and the EMIL study investigates infancy air pollution exposure.”***

The method of creating a similar score based on the one created on COPSAC data has previously been used for COPSAC replication trials to compare with different cohorts e.g. VDAART with different timepoints between the two cohorts for metabolomics (Horner, Nature Metabolism, 2025).

Results (lines 215-218) What is the mechanistic link between GI infections and air pollution? This interesting finding requires more elaboration, I think.

RESPONSE We believe that there could be different mechanisms and we have now added this to the Discussion:

Page 18, lines 330-337: ***“Current literature describes air pollutants in general to be linked with low-grade systemic inflammation, an altered gut microbiota and oxidative stress resulting in an increased risk of cardiovascular disease development in particular³³, but also leading to pathophysiological changes in the respiratory system¹² and increased risk of gastrointestinal disorders³⁴.***

Statistics: I am not an expert on PCA Analysis, maybe some other experts should be involved. Lines 230-232: PC2 Loading explains 6.5 % of the variation. Is this good loading, meaning to explain a lot of the findings, or is this association rather weak maybe? I would appreciate an explanation.

RESPONSE A PC would usually not be characterized as “weak” or “not weak”. There are more layers to it we believe. PC1 explains most of the variation, but it is not unusual to investigate the following PCs. The PCs reflect different patterns of the data. In this case, PC1 all goes in the same direction i.e. it is not useful to use this for interpretation of our data whereas PC2 reflects different directions of the proteins. For example, we have recently published a study in Nature Metabolism (Horner, 2025), where the main finding was a PC2 describing 10% of variation against neurodevelopmental outcomes because it reflected the exposure better than PC1. I hope this explanation is useful and may give a better understanding why we use PC2 and not really put much emphasis on PC1.

Reviewer #3 (Remarks to the Author):

Review of manuscript NCOMMS-25-14157. This manuscript assessed data from two cohort studies using proteomics from mother/children and assessed the associations of the proteomics linking air pollution and childhood infections. The advantages of this study is that this study has relatively large sample size and has both testing and validation

cohorts. However, I have a few questions and concerns on some methods and results interpretation of this manuscript before I can recommend accepting of this manuscript. Thanks.

RESPONSE Thank you

Major Comments:

1. In line 183, authors used Quasi-Poisson regression model to estimate the incidence rate ratio with air pollution. Do this mean the outcome is the count of each infection for each child during 0-3 years?

This analysis method will not use the longitudinal nature of the data, I wonder whether authors use this statistical method, instead of using generalized linear mixed model or time-to-event analysis?

Another limitation of this model is that the maximum count of the infection could be bounded by visit time, which may violate the model assumption. In addition, looking at Fig 1, the infection categories changed dramatically as children age increased, using a sum count of infection will loss this information.

RESPONSE Yes, the outcome is count and represents the sum of infection episodes over a 3-year period and a major strength is that we only include children having full diary follow-up data (which is a very high follow-up rate compared with other cohorts), so there are no differences in observation periods between the children analyzed. Otherwise, we could have adjusted the model for observation time, but this is not the case in this study as we restricted the analysis to only include children with full follow-up.

Using Quasi-Poisson regression is our standard approach for handling COPSAC infections data, as it accounts for overdispersion unlike a standard Poisson or time-to-event analysis, and thereby we believe that we use the strength of the longitudinal follow-up in the COPSAC cohort with a fixed follow-up time for all children to capture all episodes between 0-3 years. This is similar to what we have previously published in e.g. Bisgaard, NEJM 2016, Chawes, JAMA 2016, Brustad, JAMA Network Open, Brustad, Thorax, 2024, Kyvsgaard, JACI, 2024. By using Quasi-Poisson, we hereby have the possibility to assess children with more than one infection episode, which would be lost in a time-to-first-event analysis – which we use for e.g. asthma and eczema diagnoses.

2. In lines 205, authors mentioned 614 (88%) children had available diary data, which are the main missing factors contributing to the missing data? Authors should provide a flow chart in the Supplement showing how the participants were removed due to missingness.

RESPONSE This flowchart has now been added as an eFigure 2 in Online Supplement. The reason for not having sufficient data is almost equally divided between missing diary data of the child and insufficient maternal data.

3. For the analysis between air pollution and proteomics, did authors only used raw p-value < 0.05 as cut-off? Since there are 92 proteomic markers in the analysis, some adjustment of false discovery is needed for results presented in Fig 2.

RESPONSE No, the red line represents a $p < 0.001$ cut-off and the blue line represents $p < 0.05$. We have now made a new figure adding the FDR threshold as well as a separate figure in eFigure 4:

And added this to the Results section to emphasize that the main associations are unchanged:

Page 14, lines 240-243: ***“The false discovery rate (FDR) adjusted p-values are outlined in the Volcano plots illustrated in eFigure 4 showing similar significant associations with downregulation of AXIN1 and IL8 from PM2.5, AXIN1 from PM10 and upregulation of IL33 from NO₂ exposure.”***

4. For Fig 4, I wonder what does that mean for the y-axis (predictions) for Fig D-F?

RESPONSE The box plots represent the class predictions for the median cross-validated model. The center of the boxes represents the median, their bounds represent the 25th and 75th percentiles and the lower and upper ends of whiskers represent the smallest and largest values. This is exactly the same approach as used in our previous study (Leal, Nature Medicine, 2023). We have now added this information to the figure legend for Figure 4.

5. For Fig 5, I wonder why AXIN1 is not shown up in the PM2.5 and NO₂ signature even though AXIN1 has large loading in the sPLS regression with PM2.5 and NO₂.

RESPONSE Sorry for the confusion. It is actually included - it is called “AXI” which is now outlined for all the proteins in the legend as the names could not fit in the circles for all proteins. We did not notice this before you mentioned it. Thank you.

Page 27: ***“A correlation (Spearman) network analysis of the proteins from each air pollution component (PM_{2.5} and NO₂). The component for PM₁₀ only included 1 protein – AXIN1. Abbreviations: AXI: AXIN1, ST1: ST1A1, SIR: SIRT2 STA: STAMPB, F3L: Fit3L, CCL1: CCL11, CD4: CD40, TNF: TNFSF14, CAS: CASP8, CD8: CD8A, 4EB: 4EBP1, MMP: MMP10, IFN: IFNgamma, IL18R: IL18R1, IL1I: IL1alpha, FGF: FGF21, CSF: CSF1, IL2R: IL2RB, IL10: IL10RB”***

6. In lines 266 – 267, authors stated no fingerprints were similar between EMIL and COPSAC2010, but I did not see the fingerprints results presented for EMIL study.

RESPONSE We are unsure what the reviewer specifically requests when writing “stated no fingerprints were similar between EMIL and COPSAC2010, but I did not see the fingerprints results presented for EMIL study”. We do not claim that no fingerprints were similar – instead we are using the same model based on COPSAC data to derive a similar proteomic-pollution fingerprint in EMIL as outlined in the Methods section:

“We used the same model based on air pollution in COPSAC₂₀₁₀ to derive a similar proteomic-pollution risk score in the EMIL cohort in childhood and associated the predicted scores against infections in early childhood in both cohorts, an approach used in our previous study³⁰.”

This is a procedure we have used previously before to validate our findings in other cohorts e.g. Horner, Nature Metabolism, 2025 and is based on the score created on COPSAC air pollution data, hence, the AUC etc. are similar. This is now referenced in the manuscript.

7. This study has a testing cohort and a validation cohort, which is a plus compared to several previous studies. However, there are some discrepancies between testing analysis and validation analysis. For example, no associations between air pollution and respiratory infection in COPSAC2010 cohort, which makes the conclusion of the manuscript less convincing. Again, this might be the statistical methods used to assess associations between air pollution and childhood infection.

RESPONSE We agree that this study should not be considered as a discovery vs replication cohort, which we have now added to the Discussion and removed the terms discovery and replication throughout the manuscript:

Page 21, lines 402-405: ***Finally, the two cohorts were not directly comparable in terms of data availability and should be viewed as complementary, where the COPSAC study investigates prenatal air pollution exposure and the EMIL study investigates infancy air pollution exposure.***

Regarding the lack of overall association between air pollution and risk of infection types, we believe this is due to lack of statistical power compared with larger previous cohort studies:

Pages 17-18, lines 323-332: *“However, other larger cohort studies^{10,11,30,31,32} (n=1510, n=2568, n= 3515, n=2 199 and n=1263, respectively) have previously identified both pre- and postnatal air pollution exposure as an independent risk factor for LRTI development in young children, which could suggest that our study sample size for detecting these direct associations was too small. The ESCAPE project utilizing data from 10 birth cohorts (BAMSE, GASPII, GINIplus, LISApplus, MAAS, PIAMA and four INMA cohorts) (n=16,059) found that ambient air pollution exposure from birth increased the risk of pneumonia until age 2 years¹³. A recent (2024) large observational study (n=224,214) found ambient PM_{2.5}*

exposure in early childhood to be associated with increased risk of respiratory infections in pre-school aged children as well¹⁴."

The aim of this study is to understand the proteomic changes from air pollution and understand underlying mechanisms for the association already reported in larger cohort studies. Hence, we believe it makes sense that we do not see an overall association as the study most likely were not powered for this. The study adds to already shown associations by describing the potential underlying mechanisms with comparable proteomic results in COPSAC and EMIL.

Minor Comments:

1. The title of the manuscript seems too long. Please shorten it based on journal's requirements.

RESPONSE We agree and the title has been changed from

"Air pollution-induced maternal and child proteomic alterations increase the risk of childhood respiratory infections identifying AXIN1 protein as a potential target for asthma and respiratory infection prevention: Evidence from two European birth cohorts"

to:

"Air pollution-induced proteomic alterations increase the risk of child respiratory infections identifying AXIN1 as a potential target for prevention"

2. For Fig 5, is the link between each proteomic feature is shown as correlation? If so, please provide a legend for the color and thickness of the link for the correlation.

RESPONSE Yes the lines represent correlations. The color of the correlations is the same – green. We have now added this to the figure legend:

Page 27: "Correlations with an absolute value below 0.3 are not shown. Correlations ≥ 0.6 are plotted with maximum edge thickness of the lines. Correlations between 0.3-0.6 are scaled proportionally with thinner lines."

RESPONSE LETTER

Thank you for the invitation to resubmit our manuscript **NCOMMS-25-14157A**. We hope you will find the revised form of the manuscript of sufficient quality for publication.

REVIEWERS' COMMENTS

Reviewer #1 (Remarks to the Author):

Thanks to the authors for addressing my and other reviewers' comments. The manuscript did improve a lot, and I have no further comments.

Reviewer #3 (Remarks to the Author):

I would like to thank the authors for revising the manuscript and addressing my questions. My comments are adequately addressed. I have a few additional comments and suggestions:

1. For the sPLS-derived proteomic fingerprints of air pollution, I wonder if the prediction scores are presented in Figures 4D-F? If so, authors should make it clear that the prediction is the risk score/fingerprint.

RESPONSE This has now been added to the Figure legend 4:

*The cross-validated predictions (**fingerprints**) with AUC and p-values, which were strongly associated with air pollution. The box plots represent the class predictions for the median cross-validated model (n=613). The center of the boxes represents the median, their bounds represent the 25th and 75th percentiles and the lower and upper ends of whiskers represent the smallest and largest values.*

2. I believe the proteomic fingerprint of air pollution is a key analysis in the manuscript; the authors should provide more technical details, such as formulas and equations, or share the sPLS regression and analysis code for deriving the risk scores, thereby eliminating the need for the research community to request these details for better transparency.

RESPONSE We have now uploaded the code for deriving the risk scores to Github, which is also stated in the Code Availability statement now:

The custom code employed in this research is freely accessible to the public for transparency and reproducibility purposes at <https://github.com/tiwa1125/Air-pollution-induced-proteomic-alterations-increase-the-risk-of-child-respiratory-infections>

3. Although the authors provided explanations for the study's potential lack of power to detect air pollution (especially PM2.5 and NO2) and childhood infections, I still believe that further discussion is needed to explain the absence of associations between PM2.5/NO2 and infections, despite the proteomic fingerprint scores being associated with infections.

Thank you.

RESPONSE We have now added more about this lack of association:

Page 11, lines 200-203: ***“Another reason for not finding associations between prenatal air pollution exposure and risk of respiratory infections could be due differences in air pollution assessment, where we calculated these using different models.”***